# *Acinetobacter baumannii*: Its Clinical Significance in Human and Veterinary Medicine

**DOI:** 10.3390/pathogens10020127

**Published:** 2021-01-27

**Authors:** Francesca Paola Nocera, Anna-Rita Attili, Luisa De Martino

**Affiliations:** 1Department of Veterinary Medicine and Animal Production, University of Naples “Federico II”, 80137 Naples, Italy; francescapaola.nocera@unina.it; 2School of Biosciences and Veterinary Medicine, University of Camerino, 62024 Matelica, Italy; annarita.attili@unicam.it

**Keywords:** *Acinetobacter baumannii*, human medicine, veterinary medicine, multi-drug resistance, biofilms

## Abstract

*Acinetobacter baumannii* is a Gram-negative, opportunistic pathogen, causing severe infections difficult to treat. The *A. baumannii* infection rate has increased year by year in human medicine and it is also considered as a major cause of nosocomial infections worldwide. This bacterium, also well known for its ability to form biofilms, has a strong environmental adaptability and the characteristics of multi-drug resistance. Indeed, strains showing fully resistant profiles represent a worrisome problem in clinical therapeutic treatment. Furthermore, *A. baumannii*-associated veterinary nosocomial infections has been reported in recent literature. Particularly, carbapenem-resistant *A. baumannii* can be considered an emerging opportunistic pathogen in human medicine as well as in veterinary medicine.

## 1. Introduction

*Acinetobacter baumannii* belongs to the family Moraxellaceae and order Pseudomonadales in the class Gammaproteobacteria, phylum Proteobacteria of Eubacteria [1,2]. The development of molecular methods in the last 10 years allowed a better identification of *Acinetobacter* spp. Clinically relevant species are mostly confined to the *Acinetobacter calcoaceticus*/*Acinetobacter baumannii* (ACB) complex, namely, *A. baumannii*, *A. calcoaceticus*, *Acinetobacter pittii*, *Acinetobacter nosocomialis*, and the recently added numerous other species [3,4,5,6].

However, among the ACB complex, *A. baumannii* is the most important for its frequently involvement in hospital outbreaks, affecting critically ill and immunocompromised individuals [7,8]. A rapid and accurate identification of *A. baumannii* to distinguish it from the other species of the complex is necessary for its growing clinical interest worldwide. Among the available identification methods, matrix-assisted laser desorption/ionization time-of-flight mass spectrometry (MALDI-TOF MS) results to be the most convenient, time-saving and accurate method [9]. Furthermore, expansion and improvement of the MALDI-TOF MS database have allowed to make this instrument an efficient method for identification of *A. baumannii*, *A. pittii*, and *A. nosocomialis* [10,11]. Several molecular methods, such as whole-genome sequencing (WGS), pulsed-field electrophoresis (PFGE), multilocus variable-number tandem repeat analysis (MLVA), amplified fragment length polymorphism (AFLP) analysis, RNA spacer fingerprinting, rapid amplification of polymorphic DNA (RAPD), repetitive extragenic palindromic PCR (rep-PCR), single-locus genotyping, trilocus sequence typing (3LST), and multilocus sequence typing (MLST), are generally used for genotyping [12]. 

Genotyping studies highlighted that a limited number of clones are responsible for most of the worldwide nosocomial outbreaks [13,14]. In particular, the Global Clone 1 (GC1) and 2 (GC2) have been extensively disseminated in more than 30 countries [15]. Even though, the spread of this bacterium in the hospital setting is well known, information about its circulation outside the hospitals are still scarce [16,17]. 

It is worth remembering that for about 10 years *A. baumannii* is considered an emerging opportunistic pathogen in human medicine as also in veterinary medicine, especially for small animals such as dogs and cats [18,19]. 

Furthermore, *A. baumannii* is a member of the ESKAPE group, including six bacterial pathogens (*Enterococcus faecium*, *Staphylococcus aureus*, *Klebsiella pneumoniae*, *Acinetobacter baumannii*, *Pseudomonas aeruginosa*, and *Enterobacter* species) that are major causes of antibiotic-resistant infections and exhibiting several other virulence factors [20].

*A. baumannii* is a genus of Gram-negative bacteria, strictly aerobic, non-fermenting, non-motile, catalase-positive and oxidase-negative [7]. Multi-drug resistant (MDR) *A. baumannii* strains causing nosocomial infections with high mortality have been raising serious concerns in humans [14,21]. 

Due to its virulence properties, MDR *A. baumannii* has emerged as one of the most troublesome pathogens for health care institutions globally [22]. MDR refers to strains that exhibit resistance to three or more antimicrobial drug classes, while extensively-drug resistant (XDR) or “extreme drug resistance” strains are epidemiologically significant since they show resistance to almost all approved antimicrobial agents including carbapenems [23]. Pan-drug resistance (PDR) indicates resistance to all drug classes in addition to carbapenems also colistin, and other polymyxins [24]. However, *A. baumannii* strains show different antibiotic susceptibilities as well as considerable epidemic potential, and its rapid diagnosis is of great significance for rational use of antibiotics and shortening of treatment course.

Several virulence factors have been identified by genomic analyses, including pilus, outer membrane porins, phospholipases, proteases, lipopolysaccharides, capsular polysaccharides, protein secretion systems, and iron-chelating systems. Some strains shared genes related to great ability to adhere to cells, to invade and survive as well as to form biofilms on abiotic surface [25,26]. A recent study showed that known virulence-related factors can be present in an *A. baumannii* strain of human origin (isolated from the blood of an infected patient) even if it exhibited susceptibility to most of the antibiotics tested [27]. Additionally, it is important to remember that *A. baumannii* virulence repertoire can affect cytotoxicity, persistence, bacterial killing, and chemotaxis [28,29].

## 2. *A. baumannii* in Human Medicine

*A. baumannii* is an important and opportunistic bacterium that plays a major role in the pathogenicity in humans and predominantly infects critically ill patients. It has been isolated from human nasal samples with rates from 54 to 92% in long-term care facilities suggesting a marked tropism to the nasal mucosa [30]. Moreover, it can adhere to the surfaces in the hospital environment and survive easily for a long time in adverse conditions. However, *A. baumannii* is accountable for severe hospital-acquired infections, affecting multiple anatomical sites of patients and, in particular, those hosted in intensive care units (ICUs) [31,32,33]. Therefore, in the ICUs is very important to well define the evaluation modalities of the infection risk and the microbial ecology through an active environmental microbiological surveillance, the investigation of bacterial isolates, and the behavioral surveillance of healthcare personnel [34].

*A. baumannii* mortality rate reaches 60% in vulnerable patients [35,36]. The most clinical manifestations include pneumonia and bacteremia [37]. A survey in U.S. hospitals showed that the majority of the *A. baumannii* isolates (57.6%) were from the respiratory tract, followed by bloodstream (23.9%) and skin or wound (9.1%) in 2010 [38]. Furthermore, infections due to MDR *A. baumannii* and in particular carbapenem-resistant strains, have been associated with substantial mortality and hospital costs [39,40]. The risk factors for acquiring MDR isolates include recent exposure to antimicrobial agents, the use of venous or urinary catheters, severity of illness, duration of hospital stay, and recent surgery [41,42,43,44,45]. It has been also reported that mortality rate from invasive *A. baumannii* infection is high, especially when the isolate is resistant to carbapenems [46]. This bacterium could be transmitted through the closeness of affected patients or colonizers such as hospital surfaces and linen, and even medical equipment. In fact, contamination of respiratory support equipment, suction devices, and devices used for intravascular access are potential ways of access and transmission of infection [47]. 

Intriguingly, *A. baumannii* can be part of the bacterial microbiota of the skin. It has been demonstrated that more than 40% of healthy adults can be colonized by it on skin and mucous membranes, mainly in moist sites such as groin and toe webs; and a higher rate was observed among hospital staff members [48,49]. Trash et al. [50] have shown that, especially, the anesthesiologists can play a major role in perioperative infection control by practicing good personal hygiene and by properly disinfecting anesthetic equipment, as anesthesia providers may contribute to the risk of health-care-acquired infections.

In a recent article [51] a multivariate analysis showed that low oxygenation index and *A. baumannii* infection were risk factors for prognosis of severe influenza. Furthermore, the novel Coronavirus, known as SARS-CoV-2 responsible for Covid-19, seems to be associated with secondary bacterial infections of the lower respiratory tract of patients, more frequently due to *A. baumannii*, in particular MDR carbapemen-resistant strains [52,53,54]. These new findings highlight the concern of superinfection in patients affected by important viral infections and, consequently, this underlines the importance to limit the risk of infection and the spread of MDR strains overall in ICUs.

## 3. *A. baumannii* in Veterinary Medicine

In veterinary medicine, data regarding *A. baumannii* from animals are still scarce, even if some cases have been recently reported [55,56]. Particularly, in 2011 *A. baumannii* started to be described as an emerging pathogen in some European veterinary clinics [19,57]. Moreover, it cannot be excluded that animals may play a role as a reservoir of *A. baumannii* [58,59,60].

*A. baumannii* has been isolated from pets from various sites of infection, e.g., urinary tract infections, otitis, abscess, pneumonia, and sepsis [26,56,61,62]. However, *Acinetobacter* spp. can survive on canine healthy skin, where they may be potential reservoirs of infection, suggesting also the importance of rigorous hygiene, disinfection, and antisepsis in veterinary clinics and hospitals [61]. Belmonte et al., [58] reported pets (dogs and cats) as *A. baumannii* carriers with a prevalence of 6.5% in Reunion Island (France). Moreover, they demonstrated that most *A. baumannii* isolated strains belonged to ST25, a clonal lineage described as the one responsible for human outbreaks in Italy, Greece, and Turkey [63]. Interestingly, while the human ST25 strains appeared to be resistant to carbapenems, the veterinary isolates were susceptible to carbapenems and to almost all the tested antibiotics. It is therefore of major importance to avoid the selection and spread of MDR *A. baumannii* in animals as it is in humans, by implementation of epidemiological surveillance programs [58]. 

Interestingly, it has been reported that the cattle harbored *A. baumannii*, predominantly in the nose, and some bovine STs were the same ones already implicated in human infections; in conclusion, it appears clear that some STs are able to colonize both cattle and humans [64]. It has been also reported that the presence of *A. baumannii* in cattle shows a different prevalence among groups of the same species, such as dairy cows (21.1%), beef cattle (6.8%), and calves (2.4%). Furthermore, a seasonal occurrence was shown with a peak between May and August [64].

It is noteworthy to mention that another important public health risk has been demonstrated in a Croatian pig farm [65] with the isolation in swine manure of *A. baumannii* strains, belonging to the ST195 within GC2 which represents the predominant clone among human clinical isolates in Europe [66].

Potential reservoirs of *A. baumannii* are also habitats occupied by wildlife and livestock birds with differences between bird species and geographical areas [67]. Poultry products as turkey and chicken raw meat represent a concern since they may play a role as vehicle for transmission of MDR *A. baumannii* to humans [68].

Evidences of genotypic studies and antimicrobial drug-resistant profiles of *A. baumannii* strains isolated from hospitalized animals suggested that these microorganisms are most likely nosocomial pathogens for animals [19]. However, *A. baumannii* strains, apart from being associated with infections in hospitalized cats and dogs, have been isolated from different horses hospitalized in different stables without being associated with the illness for which the horses were hospitalized [69]. In any case, overall, the first presence of *Acinetobacter* strains showing acquired resistance to carbapenems in horses is worrying because it may also be considered a public health hazard [69]. The companion animals, such as dogs, cats and horses are the most frequently hospitalized animals and the most relevant globally with *A. baumannii*-associated diseases [19]. In any case, the role of *Acinetobacter* species in diseases of hospitalized animals is largely unknown, as well as is currently unknown the natural reservoir of *A. baumannii*. The existence of an environmental reservoir outside the healthcare settings implicated in human or animal infections cannot be excluded. 

Studies detected the presence of acquired carbapenemases in clinical isolates from pets, suggesting that they may be linked to human isolates [70,71]. Although the role of animals is still not clear in the dissemination of specific clones into the human community and hospitals, studies have demonstrated that similar or even identical *A. baumannii* clones have been identified in both settings [57]. This finding is generally limited to hospitalized animals with nosocomial infections. Thus, veterinary clinics face a great challenge regarding prevention, control, and treatment of infections with these microorganisms, similar to situations in human hospitals. It is necessary to remember that the proper hand hygiene practices, among effective control procedures, represent the key not only in human hospitals but also in veterinary hospitals. Finally, the possibility of *A. baumannii* spread from humans to animals or vice versa requires special attention. The emergence of cases of infections in companion animals associated with carbapenem-resistant isolates emphasizes the need for accurate diagnostics, because of the lack of a standard approach and specific treatment of *A. baumannii* infections in veterinary medicine. 

In addition to pets, *A. baumannii* has also been isolated from other animals with different clinical signs, including rabbit, ferret, snake, rat, and duck [72]. However, it is clear that a difference between food-producing animals, wildlife animals and companion animals could be observed, and specific clones of *A. baumannii* isolated in human are almost identical to those from companion animals, whose closer vicinity favors the transmissibility of the strains. Studies on food-producing animals reported the isolation of *A. baumannii* strains generally susceptible to antibiotics, and described for the first time three new *bla_OXA-51_* type beta-lactamase genes (*bla_OXA-148_*, *bla_OXA-149_*, and *bla_OXA-150_*) isolated from bovine *A. baumannii* strains, which have not been found previously in human isolates [73]. Differently, the isolates recovered from pig fecal samples harbored one type of *bla_OXA-51_*-like (*bla_OXA-51_* itself), which has already been reported in humans [73]. 

Particular attention deserves also cow, sheep, goat, and camel raw meat that can be sources of *A. baumannii* strains harboring antimicrobial resistance genes for both community and hospital settings environment, posing a potential risk to public health [74]. Therefore, higher levels of food-related inspection are needed, and further studies are required to define the exact role of this zoonotic pathogen in the food-chain.

## 4. Antibiotic Resistance

Antibiotic resistance is one of the biggest threats for public health. The emergence and spread of multidrug resistant (MDR) bacteria, due to increased and indiscriminate use of antibiotics, has concerned both human medicine and veterinary medicine. Generally, MDR bacterial strains are mostly associated with nosocomial infections and MDR strains are considered the cause of increased mortality rates. *A. baumannii* is well known as a major agent in nosocomial-associated infections, particularly, in ICUs of hospitals that harbor critically ill patients who are extremely vulnerable to infections. Furthermore, an increase in prevalence of MDR *Acinetobacter* spp. in hospitalized animals was also observed [19], and it was frequently diagnosed in association with infections such as urinary tract infections, abscess, pneumonia and sepsis [56,62]. 

It is worth noting that the transmission of antibiotic-resistant bacteria from animals to humans and vice versa is possible, especially in the case of potential zoonotic bacteria. In both cases, owners, operators in the veterinary sector or anyone who has contact with animals, could contract diseases or be spreaders of infections for the animals. Regarding to *A. baumannii*, strains isolated from hospitalized animals were found to be identical to those causing nosocomial infections in medical hospitals [57]. Recently we identified the same strain of *A. baumannii* in a dog and in its owner, in the first it was otitis externa-associated and in the second the strain was asymptomatically colonizing the nasal cavities [75]. 

The worldwide dissemination of MDR strains and their ability to acquire antibiotic resistance genes through several mechanisms of genetic recombination also led to the emergence of infections caused by pan-drug resistant strains (PDR), generating a worrisome therapeutic scenario [76,77]. Carbapenems are currently the antibiotics of choice against MDR *Acinetobacter* infections, but the number of *A. baumannii* strains showing resistance to carbapenems has been increasing rapidly [14] constituting the group of XDR strains. The resistance to carbapenems in *A. baumannii*, mostly mediated by acquired class D β-lactamases (mainly OXA-23; OXA-40, and OXA-58) [78], has been classified by the World Health Organization as “priority one” on the global priority list of antibiotic-resistant bacteria for research and development of new antibiotics [79].

*A. baumannii* is one of the most important MDR pathogens that, due to β-lactamases production, efflux pumps, decreased membrane permeability and altered target site of the antibiotics, is able to evade the most current therapeutic strategies. Van et al. [80] reported that more than 90% of the isolates were resistant to the tested β-lactamase/β-lactamase inhibitors, cephalosporins, carbapenems, fluoroquinolones, and trimethoprim/sulfamethoxazole, and all isolates remained sensitive to colistin. Only two years later, it was possible to read from the literature about numerous strains resistant to colistin [81,82]. In veterinary farms, colistin has been used for decades overall for the treatment and prevention of Enterobacteriaceae infections, and, in human medicine, colistin is currently considered an “antibiotic of last resort” and is used in limited amounts under strict management. However, although resistance to colistin has been described to be due to point mutations [83], the complexity of colistin resistance in *A. baumannii* is still not entirely clear [84]. 

At present, the dominant problem caused by *A. baumannii* infection is the presence of numerous MDR strains in the hospital environment, which are often insensitive to carbapenems and susceptible only to colistin [85,86]. It should be noted that the number of studies characterizing the antimicrobial resistance mechanisms of *Acinetobacter* spp. of animal origin is still very low compared with the large number of studies reporting resistance genes in human isolates. In this regard, in Table 1 are summarized papers describing MDR, XDR, and PDR *A. baumannii* strains isolated from different animal species highlighting a major spread of the first two among pets (dogs, cats, and horses), which live more closely with humans and probably even more investigated. Among poultry, cattle and pigs, wildlife animals, slaughtered animals, only in pig farm was recently detected the presence of XDR *A. baumannii* strains. Of note, in Table 1, none of the scientific papers refers to PDR *A. baumannii* strains.

Furthermore, it is well known that several and versatile *A. baumannii* virulence factors, as those responsible for adhesion, colonization and invasion, may play a role in its pathogenesis, likely contributing to its ability to survive for a long time and adapt in different environments [87,88]. Innovative strategies such as new alternative antimicrobial therapies, phage therapy and the CRISPR Cas system (Clustered regularly interspaced short palindromic repeats) have been developed to prevent the spread of MDR strains [89]. 

Amaral et al. [90] reported the growth inhibition of clinical isolates of *A. baumannii* by oregano essential oil, used alone or in combination with polymyxin B, as a promising therapeutic alternative. In fact, there are many studies on antimicrobial activity of essential oils able to inhibit the growth of both carbapenem-resistant and carbapenem-susceptible *A. baumannii* strains [91,92]. Recent advances have also demonstrated that some essential oils might increase susceptibility of conventional antimicrobial agents [93,94].

## 5. Biofilm Production

*Acinetobacter baumannii* is also well known for its ability to form biofilms, which is considered an important virulence factor. Particularly, it has been demonstrated that *A. baumannii* strains are able to form biofilms on different biotic and abiotic surfaces, so that bacteria are protected against antimicrobial and antiseptic treatments and host defenses in vivo. Indeed, the ability of *A. baumannii* to grow as biofilms may contribute to its persistence in the hospital environments, increasing the probability of nosocomial infections and outbreaks [97,98]. 

In all bacteria, biofilm formation consists of a highly organized series of molecular events influenced by different cellular and environmental signals [99]. With regard to *A. baumannii*, nutrient and iron availability, carbon sources, growth temperatures, bacterial adhesions structures (pili, flagella, outer membrane proteins, and adhesins), quorum sensing and macromolecular secretions are some of the factors influencing biofilm production [97]. Studies on the type, strain *A. baumannii* ATCC 19606T, demonstrated that biofilm initiation on abiotic surfaces is principally due to the pilus production mediated by the CsuA/BABCDE usher-chaperone assembly system. Moreover, the expression of this operon is controlled by a two-component regulatory system formed by a sensor kinase, encoded by bfmS, and a response regulator, encoded by bfmR [100,101,102].

However, the information about the interaction of *A. baumannii* with biotic surfaces such as human bronchial epithelial cells are scarce. It is known that the adhesion of this pathogen to both biotic surfaces and to plastic surfaces is associated with the bla_PER-1_ gene [103]. 

Biofilm formation ability of *A. baumannii* strains can be strengthened by the presence of the O-linked protein glycosylation system. This system, firstly described in *A. baumannii* ATCC 17978 strain, stimulates the attachment step and boosts mature biofilm mass and density [104]. Furthermore, one of the major constituents of the biofilm exopolysaccharidic matrix is represented by the poly-β-1,6-N-acetylglucosamine (PNAG). PNAG is encoded by the four genes cluster pgaABCD, harbored by almost all *A. baumannii* clinical strains [105]. 

In *A. baumannii*, as other bacterial pathogens, biofilm production represents a mechanism of resistance. Indeed, *A. baumannii* becomes metabolically inert in the deeper layers of biofilms to survive in unfavourable conditions. Poor penetration and the inability of antibiotics to act on metabolically inert bacteria increase its virulence [106]. MDR and PDR, as well as carbapenem-resistant *A. baumannii* strains, have become a relevant public’s health threat, since they have been reported with an increasing frequency worldwide [107,108,109]. Interestingly, a positive correlation between biofilm formation and antimicrobial resistance in *A. baumannii* has been confirmed [110]. In fact, it has been demonstrated that MDR clinical *A. baumannii* isolates are able to produce large quantities of biofilm [103]. However, Perez et al. [111] described an inverse relationship between biofilm production and meropenem resistance in nosocomial *A. baumannii* isolates. Therefore, the ability to form a biofilm may clearly affect antibiotic susceptibility and clinical failure, even when the dose administered is in the susceptible range [112]. Thus, *A. baumannii* strains producing biofilms could be selected under antibiotic pressure, or contrariwise, *A. baumannii* isolates might acquire multidrug-resistant profiles within biofilm communities. In both cases, the survival and circulation chances of this important pathogen in clinical settings increase [103]. Biofilm-forming *A. baumannii* are isolated especially on healthcare devices, such as urinary catheters, central venous catheters, endotracheal tubes, polycarbonate, and stainless steel [113]. 

The treatment of biofilm-associated *A. baumannii* infections represents a great challenge, since this bacterium is responsible for chronic infections. Although *A. baumannii* biofilm production mechanisms on abiotic and biotic surfaces need further studies, alternative therapeutic approaches have been studied as valid option to the classical treatment therapies, that are often unsuccessful. Natural products such as microbial, plant, and animal metabolites appeared to be effective against *A. baumannii* infections [114,115,116,117]. 

Furthermore, it is known that the metal chelator EDTA is effective against biofilms produced by *Staphylococcus* spp., *Pseudomonas* spp., and *Candida* spp. Moreover, Lee et al. [103] demonstrated that EDTA was able to reduce of 55–65% the *A. baumannii* biofilm formation. However, interestingly, Gentile et al. [118] reported that biofilm-producing *A. baumannii* was able to overcome iron restriction effected by exogenous chelators thanks to the presence of multiple iron scavenging systems. In a study it was observed that N-acetylcysteine in combination with fluoroquinolone showed an efficacy of 100% when used in vivo for the treatment of recurrent bronchopneumopathies caused by biofilm producer bacteria in dogs. The association with the mucolytic molecule improved and recovered from in vitro biofilm producer and antibiotic resistant *A. calcoaceticus* infection [119].

The ability of *A. baumannii* to produce biofilms is influenced by different factors, which play an important role in the interactions with the different abiotic and biotic surfaces. However, the mechanisms regulating adhesiveness and biofilm production vary among *A. baumannii* clinical strains and many of the molecular mechanisms are still obscure. Further studies are needed, in order to provide new information on the correlation between *A. baumanii* ability to adhere and produce biofilms, acquire multidrug-resistant profiles, and proclivity to cause chronic infections and outbreaks. The elucidation of these mechanisms is essential to achieve new therapeutic targets, leading to the introduction of new, innovative and valid antimicrobial strategies. In particular, innovative chelation-based anti-biofilm therapeutic approaches have been suggested as valid alternative approaches to ensure effective treatment options available to treat biofilm-associated *A. baumannii* infections both in human and veterinary medicine [118].

## 6. Conclusions

Antibiotic resistance is one of the biggest public health trials of our time. *A. baumannii* has developed a wide spectrum of antimicrobial resistance in comparison with non-*baumannii* species. Furthermore, antimicrobials used to treat various infectious diseases in animals may be the same or similar to those used for humans.

It is well known that one of the most effective stratagems used by bacteria to increase their survival in hostile environments is also represented by the formation of biofilms. For this reason, it is very important to find new compounds showing antimicrobial and antibiofilm properties. In recent years, several studies have been published on the latest advances in the finding of new alternatives, here mentioned. The new compounds should be studied in a transversal way given the documented diffusion of same clones in human medicine and in veterinary medicine. Thus, coming studies on clinical application of these new therapies must comply with the “One Health” approach because human health is dependent on animal health as well as ecosystem health.

The authors conclude suggesting the necessity of a high degree of attention to the possible zoonotic transmission of *A. baumannii* and to the possible human or animal infection from the environment.

## Figures and Tables

**Table 1 pathogens-10-00127-t001:** An overview and comparison of antimicrobial resistance profiles of *Acinetobacter baumannii* isolated from animals.

Geographical Origin	Animal Species	Types of Infection	MDR *	XDR **	PDR ***	References
Switzerland	Dogs, cats, and horses	Wound, pus, liver, catheter tip	x	-	-	[57]
Switzerland	Dogs and cats	Urine, abscesses, trachea, BAL, nostrils, blood	x	-	-	[18]
Reunion Island (France)	Dogs and cats	Healthy carriers (skin)	-	-	-	[58]
Germany	Dogs and cats	Hospitalized	x	-	-	[19]
Germany	Cats	Urine	x	x	-	[95]
Germany	Dogs	Skin, hairs, nostrils, throat, trachea, BAL, urine, abscesses, fistula	x	x	-	[70]
France	Dogs	Mouth, rectal swabs	-	x	-	[71]
Italy	Dogs and cats	Rectal swabs	-	x	-	[26]
Japan	Dogs and cats	Skin, pus, nasal secretions, urine, mouth mucosa, eye, vagina, ear, feces	-	-	-	[56]
UK	Dogs	Skin swabs	-	-	-	[61]
Italy	Dog	Auricular swabs	-	-	-	[75]
Belgium	Horses	Vascular catheters	x	-	-	[96]
Germany	Cattle	Fecal samples, nasal swabs, rectal swabs	-	-	-	[64]
Scotland	Cattle and pigs	Healthy carriers (fecal specimens, skin, nostril and ear swabs)	-	-	-	[73]
Croatia	Pigs	Manure	-	x	-	[65]
Germany and Poland	Chicken, geese, wild birds	Tracheal and rectal swabs	x	-	-	[67]
Iraq	Turkey and chicken	Raw meat samples	x	-	-	[68]
Iran	Cattle, sheep, goats, camels, chickens	Raw meat samples	x	-	-	[74]

Legend. MDR *: Multi-drug resistant, that means resistance to at least three class of drug (all cephalosporins, fluoroquinolones, aminoglycosides); XDR **: Extensively-drug resistant that means non-susceptible to ≥1 agent in all but ≤2 categories or MDR + resistance to carbapenems; PDR *** = Pan-drug resistant that means strains resistant to all antimicrobial agents or XDR + resistance to polymyxins; papers reporting strains with resistance to less than 3 different antimicrobial classes are those indicated by the symbol “-” in the MDR, XDR, and PDR columns.

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
