# Peer review of "Acinetobacter baumannii*: Its Clinical Significance in Human and Veterinary Medicine"

_pathogens, 2021, doi:10.3390/pathogens10020127_

Round 1

Reviewer 1 Report

This manuscript reviews studies concerning both human and animal Acinetobacter baumannii infections. However, they fail the main goal of the work due to the lack of organized ideas, but also by failing in presenting  the previous studies with more detail. For instance, a table presenting the animal infections described until now, with information regarding the main characteristics of the isolates would be an important addition to the manuscript.

I also have concerns regarding the references list. A number of old references regarding the taxonomy history of the Acinetobacter genus were used (e.g., 1-8), but this is a theme already explored in previous reviews and not the focus of this manuscript. On the other hand, a number of reviews were used when the original studies should have been used. 

Another important issue is the confusion regarding the species identification: 1) ACB complex composition - in the second paragraph is referred that the group is constituted by A. baumannii, A. calcoaceticus, Acinetobacter genomic species 3, and Acinetobacter genomic species 13 (here should be 13 TU), but in the third paragraph the authors used the current species designation without referring the correspondence (Acinetobacter pittii for Acinetobacter genomic species 3, and Acinetobacter nosocomialis for Acinetobacter genomic species 13); 2) consider that infection was caused by A. baumannii when the original work refers only to Acinetobacter (e.g., ref 53).

Finally, the authors do not have a record on studies regarding Acinetobacter, which may also justify the drawbacks observed in the manuscript.

Author Response

Dear Reviewer,

Herein, we submit a revised form of the review entitled “Acinetobacter baumannii: Its Clinical Significance in Human and Veterinary Medicine” n.1049948 by Nocera et al., modified according to the reviewer’s suggestions, and a response point by point to your comments. All the modified sentences are displayed in the revised manuscript with yellow marked changes.

Reviewer 1 - Comments and Suggestions for Authors

This manuscript reviews studies concerning both human and animal Acinetobacter baumannii infections. However, they fail the main goal of the work due to the lack of organized ideas, but also by failing in presenting the previous studies with more detail. For instance, a table presenting the animal infections described until now, with information regarding the main characteristics of the isolates would be an important addition to the manuscript.

- Thank you for your suggestions. We revised all the manuscript and added a table presenting the animal infections caused by A. baumannii. We revised also all the references. A revision of the English was done throughout the manuscript.

I also have concerns regarding the references list. A number of old references regarding the taxonomy history of the Acinetobacter genus were used (e.g., 1-8), but this is a theme already explored in previous reviews and not the focus of this manuscript. On the other hand, a number of reviews were used when the original studies should have been used. 

- We revised all the references on taxonomy and deleted the oldest references.

Another important issue is the confusion regarding the species identification: 1) ACB complex composition - in the second paragraph is referred that the group is constituted by A. baumannii, A. calcoaceticus, Acinetobacter genomic species 3, and Acinetobacter genomic species 13 (here should be 13 TU), but in the third paragraph the authors used the current species designation without referring the correspondence (Acinetobacter pittii for Acinetobacter genomic species 3, and Acinetobacter nosocomialis for Acinetobacter genomic species 13);

 - We revised and shortened all the paragraphs concerning the species identification.

2) consider that infection was caused by A. baumannii when the original work refers only to Acinetobacter (e.g., ref 53).

- We revised it by adding proper references. Thank you.

Finally, the authors do not have a record on studies regarding Acinetobacter, which may also justify the drawbacks observed in the manuscript.

Reviewer 2 Report

In the manuscript in question the authors are reviewing the literature on Acinetobacter baumannii and its clinical significance in human and veterinary medicine. The manuscript is of interest. Below are some suggestions to improve it:

Line 68: Change “Thanks” to “Due”.

Lines 83-86: Shorten sentence. Also, what is a severe host environment?

Line 89: Change to “pathogenicity in humans”.

Line 90: I think the authors mean “long term care facilities”

Line 94: Intensive care unit is usually abbreviated as ICU. This abbreviation is used at a later stage in the manuscript.

Lines 94-97: I recommend omitting the sentence. If not, change “infectious risk” to “infection risk”.

Line 120: Change “responsible of” to “responsible for”.

Line 134: Change “to be” to “being”.

Line 137: Change “first description” to “presence”.

Line 138: Change to “Companion animals such as dogs, cats and horses” if applicable.

Line 141: Change to “as is the natural reservoir of A. baumannii”.

Line 143: Which bacterial strain?

Line 163: Change to “other animals”

Line 192: Are mortality rates increasing or are they increased because of MDR bacterial strains?

Line 193: Use either the words healthcare-associated or nosocomial, but not both.

Lines 204-205: Change to “asymptomatically colonizing the nasal cavities”.

Lines 206-208: Change to: “The worldwide dissemination of MDR strains and their ability to acquire antibiotic resistance genes through several mechanisms of genetic recombination also led to the emergence of infections caused by pandrug-resistant strains, generating a worrisome therapeutic scenario.”

Lines 209-212: I recommend omission of the two sentences.

Line 221: Change “survive to the" to “evade”.

Line 225: Omit “suggesting the rational and necessary use of this antibiotic”

Lines 228-230: Change sentence to “However, although resistance to colistin has been described to be due to point mutations, the complexity of colistin resistance in A. baumannii is still not entirely clear”.

Lines 239-241: Change sentence to: “Innovative strategies such as new alternative antimicrobial therapies, phage therapy and the CRISPR Cas system (Clustered regularly interspaced short palindromic repeats) have been developed to prevent the spread of MDR strains.”

Line 244: Change “However” to “in fact”.

Line 249: Change “that” to “which”.

Line 252: Change to “the ability of A. baumannii”.

Line 264: Change “Regarding to” “With regard to”

Line 267: Remove “Precisely,”

Line 268: Remove the comma.

Line 272: Change “Differently” to “However”.

Line 274: Change “mediated by” to “associated with” as the blaPER-1 gene may not be the cause of biofilm formation.

Line 279: change “whilst plays a partial role” to “and plays a role”.

Line 310: Remove “Precisely,”

Line 328: Change A. baumannii ability” to “the ability of A. baumannii

Author Response

Dear Reviewer,

Herein, we submit a revised form of the review entitled “Acinetobacter baumannii: Its Clinical Significance in Human and Veterinary Medicine” n.1049948 by Nocera et al., modified according to the reviewer’s suggestions, and a response point by point to your comments. All the modified sentences are displayed in the revised manuscript with yellow marked changes.

Reviewer 2 - Comments and Suggestions for Authors

In the manuscript in question the authors are reviewing the literature on Acinetobacter baumannii and its clinical significance in human and veterinary medicine. The manuscript is of interest. Below are some suggestions to improve it:

Line 68: Change “Thanks” to “Due”.

- Thank you, for your suggestion. We made it.

Lines 83-86: Shorten sentence. Also, what is a severe host environment?

- We did it.

Line 89: Change to “pathogenicity in humans”.

- Thank you, we changed it.

Line 90: I think the authors mean “long term care facilities”

- Yes. We corrected it.

Line 94: Intensive care unit is usually abbreviated as ICU. This abbreviation is used at a later stage in the manuscript.

- Thank you. We corrected it.

Lines 94-97: I recommend omitting the sentence. If not, change “infectious risk” to “infection risk”.

- We decided to leave this part and to change with infection risk.

Line 120: Change “responsible of” to “responsible for”.

- Thank you, we did it.

Line 134: Change “to be” to “being”.

- We made this correction. Thank you.

Line 137: Change “first description” to “presence”.

- We did it.

Line 138: Change to “Companion animals such as dogs, cats and horses” if applicable.

- We corrected it.

Line 141: Change to “as is the natural reservoir of A. baumannii”.

- We made this correction.

Line 143: Which bacterial strain?

- Thank you for your suggestion, we talked about of A. baumannii. We defined it

Line 163: Change to “other animals”

- We corrected it.

Line 192: Are mortality rates increasing or are they increased because of MDR bacterial strains?

- We talked about MDR bacterial strains as a cause of an increase of the mortality rates. We changed the sentence as follows: “MDR strains are considered the cause of increased mortality rates”.

Line 193: Use either the words healthcare-associated or nosocomial, but not both.

- Thank you, we corrected it by using only “nosocomial-associated infections.”

Lines 204-205: Change to “asymptomatically colonizing the nasal cavities”.

- Thank you. We changed it.

Lines 206-208: Change to: “The worldwide dissemination of MDR strains and their ability to acquire antibiotic resistance genes through several mechanisms of genetic recombination also led to the emergence of infections caused by pandrug-resistant strains, generating a worrisome therapeutic scenario.”

- Thank you. We made as you suggested. 

Lines 209-212: I recommend omission of the two sentences.

- We did it.

Line 221: Change “survive to the" to “evade”.

- We made this change.

Line 225: Omit “suggesting the rational and necessary use of this antibiotic”

- Thank you, we did it.

Lines 228-230: Change sentence to “However, although resistance to colistin has been described to be due to point mutations, the complexity of colistin resistance in A. baumannii is still not entirely clear”.

- We did as you suggested, thank you.

Lines 239-241: Change sentence to: “Innovative strategies such as new alternative antimicrobial therapies, phage therapy and the CRISPR Cas system (Clustered regularly interspaced short palindromic repeats) have been developed to prevent the spread of MDR strains.”

- Thank you, we changed the sentenced as you suggested.

Line 244: Change “However” to “in fact”.

- We did it. Thank you.

Line 249: Change “that” to “which”.

- We changed it.

Line 252: Change to “the ability of A. baumannii”.

- We did it.

Line 264: Change “Regarding to” “With regard to”

- We corrected it.

Line 267: Remove “Precisely,”

- Done.

Line 268: Remove the comma.

- Done.

Line 272: Change “Differently” to “However”.

- We changed it.

Line 274: Change “mediated by” to “associated with” as the blaPER-1 gene may not be the cause of biofilm formation.

- We did it.

Line 279: change “whilst plays a partial role” to “and plays a role”.

- We corrected it.

Line 310: Remove “Precisely,”

- Done.

Line 328: Change “A. baumannii ability” to “the ability of A. baumannii

- We did as you suggested.

Reviewer 3 Report

Nocera FP. “Acinetobacter baumannii: Its clinical significance in human and veterinary medicine” (Pathogens-1049948)

This review introduced clinical importance of A. baumannii in human and veterinary medicine.

  1. So far, much has been reviewed on the clinical importance of A. baumannii in humans, but this review is meaningful because it is relatively rare in the field of veterinary medicine. I think it will be a better review if the authors can reinforce the relationship between A. baumannii in humans and animals.

  1. It is worth that new data in COVID-19 era are included in this review. It would be better if the authors could further investigate if there are more papers on this.

  1. “Antibiotic resistance”. This part mainly deals with antibiotic resistance of A. baumannii in humans, and needs to be reinforced in the animal field.

  1. “Biofilm production”. It is not well understood why the biofilm section is covered in depth. It is not generally described as an important trait in A. baumannii, and there is no explanation as to why this feature is more important than other features as a virulence factor.

  1. Many sentences are in error and should be checked carefully.

  1. Line 31-39. It is necessary to mention which genomic species has been newly characterized. For example, Acinetobacter genomic species 13TU was newly characterized as A. seifertii.

  1. Line 54. “the international clones 1 and 2”. There is a generic term for these, Global Clone 1 and 2 (GC1) and GC2.

Author Response

Dear Reviewer,

Herein, we submit a revised form of the review entitled “Acinetobacter baumannii: Its Clinical Significance in Human and Veterinary Medicine” n.1049948 by Nocera et al., modified according to the reviewer’s suggestions, and a response point by point to your comments. All the modified sentences are displayed in the revised manuscript with yellow marked changes.

Reviewer 3 - Comments and Suggestions for Authors

Nocera FP. “Acinetobacter baumannii: Its clinical significance in human and veterinary medicine” (Pathogens-1049948)

 This review introduced clinical importance of A. baumannii in human and veterinary medicine. 

  • So far, much has been reviewed on the clinical importance of A. baumannii in humans, but this review is meaningful because it is relatively rare in the field of veterinary medicine. I think it will be a better review if the authors can reinforce the relationship between A. baumannii in humans and animals.

- Thank you for your suggestion, we reinforced the relationship between A.baumannii in humans and animals as you requested.

  • It is worth that new data in COVID-19 era are included in this review. It would be better if the authors could further investigate if there are more papers on this.

-We added new other papers about this item.

  • “Antibiotic resistance”. This part mainly deals with antibiotic resistance of A. baumannii in humans, and needs to be reinforced in the animal field.

- We did it.

  • “Biofilm production”. It is not well understood why the biofilm section is covered in depth. It is not generally described as an important trait in A. baumannii, and there is no explanation as to why this feature is more important than other features as a virulence factor.

- We shortened this section as requested.

  • Many sentences are in error and should be checked carefully.

- We checked all the sentences. 

  • Line 31-39. It is necessary to mention which genomic species has been newly characterized. For example, Acinetobacter genomic species 13TU was newly characterized as A. seifertii.

- We revised and corrected the species identification. Moreover, we shortened the taxonomy information as suggested by the reviewer 1.

  • Line 54. “the international clones 1 and 2”. There is a generic term for these, Global Clone 1 and 2 (GC1) and GC2.

- We changed as you suggested.

Round 2

Reviewer 3 Report

.